# Development of a simple screening tool for determining cognitive status in Alzheimer's disease

Hsin-Te Chang[1,2], Pai-Yi Chiu [ID][3,4]*

1 Department of Psychology, College of Science, Chung Yuan Christian University, Taoyuan, Taiwan,
2 Research Assistance Center, Show Chwan Memorial Hospital, Changhua City, Taiwan, 3 Department of Neurology, Show Chwan Memorial Hospital, Changhua City, Taiwan, 4 Department of Applied Mathematics, College of Science, Tunghai University, Taichung, Taiwan

* paiyibox@gmail.com

**Data Availability Statement:** All de-identified raw data are available from the BioStudies Submission Tool database (accession number: S-BSST951).

**Funding:** The authors received no specific funding for this work.

## Abstract

Cognitive screening is often a first step to document cognitive status of patients suspected having Alzheimer's disease (AD). Unfortunately, screening neuropsychological tests are often insensitivity in the detection. The goal of this study was to develop a simple and sensitive screening neuropsychological test to facilitate early detection of AD. This study recruited 761 elderly individuals suspected of having AD and presenting various cognitive statuses (mean age: 77.69 ± 8.45 years; proportion of females: 65%; cognitively unimpaired, CU, n = 133; mild cognitive impairment, MCI, n = 231; dementia of Alzheimer's type, DAT, n = 397). This study developed a novel screening neuropsychological test incorporating assessments of the core memory deficits typical of early AD and an interview on memory function with an informant. The proposed History-based Artificial Intelligence-Show Chwan Assessment of Cognition (HAI-SAC) was assessed in terms of psychometric properties, test time, and discriminative ability. The results were compared with those obtained using other common screening tests, including Cognitive Abilities Screening Instrument (CASI), Montreal Cognitive Assessment (MoCA), and an extracted Mini-Mental State Examination score from CASI. HAI-SAC demonstrated acceptable internal consistency. Factor analysis revealed two factors: memory (semantic and contextual) and cognition-related information from informants. The assessment performance of HAI-SAC was strongly correlated with that of the common screening neuropsychological tests addressed in this study. HAI-SAC outperformed the other tests in differentiating CU individuals from patients with MCI (sensitivity: 0.87; specificity: 0.58; area under the curve [AUC]: 0.78) or DAT (sensitivity: 0.99; specificity: 0.89; AUC: 0.98). Performance of HAI-SAC on differentiating MCI from DAT was on par with performances of other tests (sensitivity: 0.78; specificity: 0.84; AUC: 0.87), while the test time was less than one quarter that of CASI and half that of MoCA. HAI-SAC is psychometrically sound, cost-effective, and sensitive in discriminating the cognitive status of AD.

**Competing interests:** The authors have declared that no competing interests exist.

## Introduction

Aging and Alzheimer's disease (AD) are global challenges [1]. Early and effective diagnosis of AD is a critical public health issue. Cognitive examinations are critical in diagnosing dementia of the Alzheimer's type (DAT) and mild cognitive impairment (MCI) [2–4]. Differentiating among cognitive statuses in AD generally requires a comprehensive neuropsychological assessment; however, performances on neuropsychological assessments and daily functioning may be inconsistent [5]. In addition, administering comprehensive assessments is arduous and time-consuming, which may hinder the recruitment of patients for clinical trials [6]. Various screening neuropsychological tests have been developed for the purpose of identifying individuals in need of comprehensive assessment [7, 8]. Unfortunately, although screening neuropsychological tests have been shown to provide high specificity, they suffer from low sensitivity, especially in detecting early AD-related cognitive changes [9].

Previous studies have pointed out the importance of reports from patient informants in diagnosing cognitive status and predicting cognitive decline in AD [10–12]. In detecting cognitive impairments, recent research has emphasized behavioral indices that are rely on informants providing patient support (e.g., Attended With and Head Turning Signs) [13]. Researchers have demonstrated that a combination of objective performances on cognitive tasks and information related to daily living functions gathered from informant may be more sensitive in detecting cognitive changes in early stages of AD [12, 14]. The advantage of the combination of objective performance of the patient and information reported by informant on cognitive functions and daily living functions may lie in the assessment of impact of cognitive decline on daily life, which is crucial in determining cognitive status of the individual [12, 14].

The first cognitive symptom of AD is usually memory dysfunction [2, 15–17], particularly contextual binding [18–20] and semantic memory [21–23]. Existing screening neuropsychological tests often assess global cognition, thereby limiting ability to detect early cognitive changes by diluting effects of cognitive deficit on overall cognitive status [21, 24].

Limited screening neuropsychological tests have been validated in Taiwan [25–28]. This study aimed to develop a new and simple screening test, History-Based Artificial Intelligence-Show Chwan Assessment of Cognition (HAI-SAC), for the assessment of cognitive status in Alzheimer's disease which incorporate objective cognitive functions and information related to daily living of the patients provided by the informant. To our knowledge, this is the first study to incorporate informants' reports with a focus on early cognitive changes in DAT with a screening neuropsychological test. This study included cognitively unimpaired elderly individuals (CU) and patient with MCI, or DAT. Psychometric properties of the HAI-SAC were evaluated. We also compared HAI-SAC and conventional screening neuropsychological tests used in Taiwan in their ability to determine cognitive status of AD patients. We hypothesized that 1. The HAI-SAC would have good psychometric properties; 2. The HAI-SAC would outperform traditional screening neuropsychological tests in discriminating cognitive status of AD patients.

## Materials and methods

### Participants

This is a substudy of "History-Based Artificial Intelligence Clinical Dementia Diagnostic System Project (HAICDDS)" with a retrospective analysis of a dementia registry database from Show Chwan Healthcare System, which is currently applied in three centers in Taiwan [26, 29, 30]. In this database, Clinical Dementia Rating (CDR) [31] scores were obtained for staging of

dementia and daily function was based on Instrumental Activities of Daily Living (IADL) Scale [32]. Cognitive function was assessed using Cognitive Abilities Screening Instrument (CASI) [7] and Montreal Cognitive Assessment (MoCA) [33]. Neuropsychiatric symptoms were evaluated using Neuropsychiatric Inventory (NPI) [34]. All participants underwent CASI and MoCA assessments by trained neuropsychologists. In performing CDR and IADL, informants were also interviewed by trained neuropsychologists. Vascular risk factors (hypertension, diabetes, hypercholesterolemia, coronary heart disease) were identified by reviewing medical charts or structured interviews conducted by neuropsychologists. Diagnoses of subtype of dementia of cognitive impairment and its severity by neurologists after obtaining all clinical data, including medical history, neuropsychological tests, brain imaging, and laboratory data. Undetermined cases were discussed in consensus meetings. This study recruited a total of 761 participants aged over 60 years with CU (n = 133), MCI due to AD (n = 231), or DAT (n = 397). Data from the HAICDDS database were analyzed retrospectively and anonymously. The study was approved by the Committee for Medical Research Ethics of Show Chwan Memorial Hospital and informed consent was waived (SCMH_IRB No: IRB1081006).

## Diagnosis of CU, MCI or dementia

Cognitive statuses of the participants were identified through regular consensus meetings aimed at gathering the opinions of neurologists and clinical neuropsychologists. The cognitive function of participants was evaluated by clinical neuropsychologists. Daily function was assessed by neurologists or clinical neuropsychologists in accordance the IADL [14, 32]. CU was defined as individuals with a global CDR score of 0 without subjective cognitive complaints or of 0.5 with complaints only in the memory domain and testing results (CASI) in the normal range [25]. Diagnosis of MCI due to AD was based on criteria outlined by National Institute on Aging and Alzheimer's Association (NIA-AA) Workgroup [3]: Changes in cognition with impairments in cognitive tests (CASI) [25], but without evidence of social or occupational dysfunctions (IADL > 6) with a CDR score of 0.5 [14]. Cutoff scores of MCI/dementia using CASI were in the non-demented range adjusted for age and education [25]. Diagnosis of probable AD dementia was based on criteria put forth by NIA-AA [2]. Informants were interviewed individually and separately from patients.

## CASI

The CASI is a globally recognized method of measuring cognition which has been used in a number of international studies of dementia [7]. CASI assesses long-term memory, by asking the subjects about general knowledge (e.g., "How many months are there in a year?") (10 points); short-term memory, using learning a list of three nouns and a short delayed recall trial approximately 5 minutes after the learning (9 points), a long delayed recall trial approximately 20 minutes after the learning (9 points), and a recognition task after learning of five common objects (5 points) (the short term memory score is calculated using the following formula: 1/2 [score of short delayed recall + score of long delayed recall] + 0.6 x score of recognition); orientation, by asking the participants about their age (2 points), the date (9 points), day of the week (1 point), time of the day (1 point), and orientation to space (5 points); attention, using the performance on the above mentioned learning trial (6 points) and repetition of two syntactically complex sentences (2 points); mental manipulation, using a digit backward task (5 points) and a serial subtraction task (5 points); abstract thinking, by asking the subjects to determine the similarities between two concepts and to provide solutions under situations that may happen in daily life (e.g., "What will you do if you lose an umbrella that was borrowed from other?") (6 points); language, by asking the subjects to perform an action that is indicated by a

written sentence (3 points), to write five Chinese characters with high word frequency (5 points), a ten-item confrontation naming task with body parts (forehead, jaw, shoulder, palm, and thumb) and common items (spoon, coin, toothbrush, key, hairbrush) (10 points), and a task asking the subject to following commands (3 points) (the language score is calculated using the following formula: 1/2[score on the reading comprehension task + score on the writing task] + 0.3 x score on the confrontational naming task + score on the following commands task); visuospatial abilities, using a pentagons drawing task (10 points); and verbal fluency using an animal category (10 points). The scores of CASI range from 0 to 100. CASI has been adapted among Taiwanese individuals. The cutoff scores of CASI for screening of dementia are stratified according to age, sex, and educational levels, range from 46/47 for illiterate individuals aged over 80 years to 81/82 for individuals aged over 80 years with educational levels higher than elementary school (6 years) [25].

## MoCA

The MoCA has been widely used across various countries for detecting cognitive dysfunction which assess short-term memory, using two learning trials of five nouns and delayed recall after approximately 5 minutes (5 points); visuospatial abilities, using a clock-drawing task (3 points) and a three-dimensional cube copy (1 point); executive functions, using the Trail Making Test-Part B (TMT-B)(1 point), a verbal fluency task, and a two-item verbal subtraction task (2 points); attention, using a sustained attention task (target detection using tapping, 1 point), a serial subtraction task (3 points), and digits forward and backward (1 point each); language using a three-item confrontation naming task with low-familiarity animals (lion, camel, rhinoceros; 3 points), repetition of two syntactically complex sentences, and the verbal fluency; and orientation to time and place (6 points) [8]. Items on the Taiwanese version were identical to the English version with the exception of the following five cultural and linguistic modifications: the TMT-B were modified by using Chinese nominal sequential words instead of English alphabet [35]; the items of sentence repetition were replaced with Chinese sentences; a fruit category fluency task replaced the phonemic letter fluency, as there are no letter-equivalent linguistic units in the Chinese language; district was substituted for city for the assessment of orientation. In Taiwan, districts are the conceptual equivalent of cities in North American countries. The cutoff score for MoCA for screening of dementia is 23/24 in Taiwan [36, 37].

## NPI

The NPI was developed to assess the neuropsychiatric symptoms that occur in AD and other neurodegenerative diseases [34]. We used the 12-item version with caregiver distress that have previously validated in Taiwan assessing symptoms of delusion, hallucination, agitation or aggression, depression, anxiety, euphoria, apathy, disinhibition, irritability, motor disturbance, nighttime behaviors, and appetite and eating behavior changes reported by the informant [38, 39]. A sum of boxes of NPI (NPI-SB) score was obtained by calculating the results of multiplication of frequency (0 to 4, from 0: none to 4: very frequent), severity (0 to 3, from 0: none to 3: severe), and distress (0 to 5, from 0: none to 5: extremely distressed).

## IADL

Informants were asked about participant's ability during the previous month to perform eight activities using the Lawton's IADL [14, 32]: shopping, transportation, finances, telephone use, medication, food preparation, household chore, and laundry. One point is assigned if the participant is deemed independent in each of the activities. A summary score ranges from 0 (low function: dependency) to 8 (high function: independent).

## HAI-SAC

The items of HAI-SAC were selected from consensus meetings gathering opinion from a senior neurologist (PYC) and a clinical neuropsychologist (HTC) and were designed according to the findings in terms of early cognitive changes among patients with DAT [20, 40]. HAI-SAC comprising 3 objective cognitive tests by which to evaluate semantic and contextual memory functioning and two informant-based questions. Stimuli used in the cognitive tests were intentionally selected using objects that may be frequently encountered by people lived in Taiwan (umbrella, glasses, mobile phone, key, watch, and kettle) [41]. The cognitive tests include following: 1. Confrontational naming of objects, colors, and details (CNOCD): Participants are provided photos depicting individual objects arranged with fixed sequence (i.e., umbrella, glasses, mobile phone, key, watch, and kettle from left to the right relative to the subject). Participants are required to name each of the objects ("What is the name of this?"), their main color (e.g., "What color is this umbrella?"), and details of the object (i.e., "What time is shown in this watch?"); 2. Memory for location (ML): Participants are tasked with arranging the photos as previously shown in CNOCD; and 3. Recalling functional properties of objects (RFO): Participants are tasked with recalling two functions of each object with photos arranged in sequence used in CNOCD. CNOCD and RFO are meant to place a heavy load on semantic memory functioning [40], and ML is meant to place cognitive load on contextual binding abilities [20]. The two informant-based questions deal with 1. Cognitive decline (CD), in which informant is asked whether participants suffer from cognitive decline, and 2. Functional decline (FD), in which informant is asked whether the observed decline in cognitive function interferes with daily activities of participant. A score of 1 point is assigned for each correct response in CNOCD, except for questions related to details, which earn scores as follows: correct time (4 points); correct hour and minute but incorrect second (2 points); correct minute but incorrect second (1 point). Two points are awarded for each photo that is located correctly during the ML. An additional two points are awarded for a perfect response. Negative responses for CD questions are scored 8 points and negative responses for FD questions are scored 12 points. Categorization of CU, MCI, or DAT is performed independently from the HAI-SAC evaluation.

## Statistical analyses

Statistical analyses were performed using SPSS 15.0 for Windows (IBM, Chicago) and SAS (SAS Institute Inc., Cary). Demographics, neuropsychological tests, CDR-sum of boxes (CDR-SB), IADL, CASI, MoCA, NPI-SB, and test time were compared among groups using one-way ANOVA with Bonferroni corrections (with $\alpha$ levels set at 0.017). Gender ratios among groups were analyzed using chi-square tests. Internal consistency of items was examined by calculating Cronbach's $\alpha$. Exploratory factor analysis (EFA) was performed on HAI-SAC using oblique factor rotation (Promax method) and principal axis extraction [42]. Suitability of EFA to data was examined using Kaiser-Meyer-Olkin (KMO) test for sampling adequacy (less than 0.6 indicates inadequacy) and Barlett's test for sphericity (acceptance of null hypothesis indicates inadequacy) [43]. Scree plots and levels of variance were used to determine number and patterns of factors. Items with factor loading exceeding 0.30 on a given factor were used for interpretation. Factor scores were calculated by a summation of each factor loading by score on each of the item. Correlation between scores on factor scores of HAI-SAC and scores on the conventional screening neuropsychological tests including a extracted MMSE score from CASI [7], CDR-SB, NPI-SB, and IADL were evaluated using Pearson correlation coefficients. The strengths between HAI-SAC scores and scores on other tests were assessed in accordance with previous researches: absolute values higher than 0.4 indicate

moderate correlations and absolute values higher than 0.7 indicate strong correlations [44]. Discriminant analysis comparing accuracy in categorization of CU, MCI, or DAT using CASI, MoCA, extracted MMSE, and HAI-SAC was based on receiver operating curve (ROC) analysis. We separated objective cognitive (HAI-SAC-3) and informant-based assessments (HAI-SAC-I) to investigate relative contributions of individual HAI-SAC components in discriminations. Youden's index was used to determine optimal cut-off scores for differentiation among cognitive stages (maximum = sensitivity + specificity– 1). The areas under the curves (AUCs) calculated using performances on each test were compared using methods suggested in previous studies using proc logistic program in SAS [45].

## Results

### Demographical and clinical characteristics

Table 1 lists demographic and clinical characteristics in the various groups. The mean age of participants was 77.69 ± 8.45 years (range: 60–100 years) and the mean education level was 5.27 ± 4.68 years (range: 0–23 years). The proportion of females was 65%. Patients in DAT and MCI groups were older than those in CU group, and DAT patients were older than MCI patients ($F_{(2,758)}$ = 92.97, $p < 0.001$). Educational level of individuals in CU group was higher than in MCI and DAT groups, and educational level in MCI group was higher than in DAT group ($F_{(2,758)}$ = 25.45, $p < 0.001$). DAT group included a higher percentage of females than did CU and MCI groups ($\chi^2_{(d=2,\ n=761)}$ = 10.78, $p < 0.01$). Scores on the CDR-SB and NPI-SB were higher in the DAT group than score in the MCI and CU groups, and scores on the CDR-SB and NPI-SB were higher in the MCI than CU groups (CDRSB, $F_{(2,758)}$ = 266.11, $p < 0.001$; NPISB, $F_{(2,1\ 758)}$ = 11.79, $p < 0.001$). Scores on the IADL in the CU group were higher than scores in the MCI and DAT groups, and scores on the IADL in the MCI group were higher than scores in the DAT group ($F_{(2,758)}$ = 487.55, $p < 0.001$). No between-group differences were observed in the proportion of individuals with hypertension, diabetes, or coronary heart disease. The number of individuals with

**Table 1. Demographic and clinical characteristics across groups.**

|  | CU | MCI | DAT | Statistical significance |
|---|---|---|---|---|
| N | 133 | 231 | 397 |  |
| Age | 72.15 (7.38)[ab] | 74.90 (7.29)[ac] | 81.17 (7.81)[bc] | $p < 0.001$ |
| Education | 7.38 (4.76)[ab] | 5.78 (4.43)[ac] | 4.27 (4.53)[bc] | $p < 0.001$ |
| Gender (% male) | 44.40 (59/133)[b] | 38.50 (89/231)[c] | 30.00 (119/397)[bc] | $p < 0.001$ |
| CDR-SB (maximum = 18) | 0.35 (0.32)[ab] | 1.96 (2.16)[ac] | 7.17 (4.58)[bc] | $p < 0.001$ |
| NPI-SB (maximum = 144) | 2.64 (3.45)[ab] | 4.42 (6.06)[ac] | 6.16 (9.18)[bc] | $p < 0.001$ |
| IADL (maximum = 8) | 7.79 (0.55)[ab] | 6.58 (2.03)[ac] | 2.20 (2.53)[bc] | $p < 0.001$ |
| Hypertension (%) | 56.39 (75/133) | 58.01 (134/231) | 55.67 (221/397) | $p = 0.85$ |
| Diabetes (%) | 27.07 (36/133) | 32.90 (76/231) | 27.96 (111/397) | $p = 0.35$ |
| Hypercholesterolemia (%) | 39.10 (52/133)[b] | 35.93 (83/231)[c] | 22.17 (88/397)[bc] | $p < 0.001$ |
| Coronary heart disease (%) | 4.51 (6/133) | 6.49 (15/231) | 5.04 (20/397) | $p = 0.65$ |

[a] CU ≠ MCI;

[b] CU ≠ DAT;

[c] MCI ≠ DAT.

CDR-SB = Clinical Dementia Rating-sum of boxes; CU: Cognitive unimpaired; DAT: Dementia of the Alzheimer's type; IADL: Instrumental activities of daily living; MCI: Mild cognitive impairment; NPI-SB: Neuropsychiatric Inventory-sum of boxes.

**Table 2. Correlations between HAI-SAC and its components and other clinical assessments.**

|  | MMSE | CASI | MoCA | CDR-SB | NPI-SB | IADL |
|---|---|---|---|---|---|---|
| HAI-SAC | 0.81 | 0.86 | 0.81 | -0.81 | -0.15 | 0.80 |
| SMCB | 0.82 | 0.89 | 0.78 | -0.81 | -0.13 | 0.70 |
| INF | 0.46 | 0.47 | 0.51 | -0.49 | -0.11 | 0.55 |

All *p* values < 0.005. CASI: Cognitive Assessment Screening Instrument; CDR-SB: Clinical Dementia Rating-Sum of Boxes; HAI-SAC: History-Based Artificial Intelligence-Show Chwan Assessment of Cognition; IADL: Lawton's Instrumental Activities of Daily Living; INF: Cognitive-related information derived from informants; MMSE: Mini-Mental State Examination; MoCA: Montreal Cognitive Assessment; NPI-SB: Neuropsychiatric Inventory-sum of boxes; SMCB: Semantic memory and contextual binding.

hypercholesterolemia in the CU (39.10%) and MCI (35.93%) groups exceeded that in the dementia group (22.17%).

## Internal consistency and EFA

Internal consistency of items in HAI-SAC was acceptable (Cronbach's $\alpha$ = 0.63). KMO (0.74) and Bartlett's tests ($\chi^2_{df=10,\ n=761} = 1474.48, p < 0.001$) indicated the data were suitable for EFA. Two of components could be used to interpret scores in HAI-SAC: First component, explaining 55.23% of variance, dealt with semantic memory (CNOCD, factor loading: 0.95; RFO: 0.96) and contextual binding ability (ML: 0.55) (SMCB). Second component, explaining 19.91% of variance, dealt with cognitive-related information derived from informants (CD: 0.97; FD: 0.45) (INF). FD cross loaded on both components (0.48 on semantic memory and contextual binding ability).

## Criterion-related validity

Performances on the HAI-SAC and SMCB strongly correlated with performances on the MMSE, CASI, MoCA, CDR-SB, and IADL (|*r*| = 0.70 ~ 0.89). Performances on the INF moderately correlated with performances on the MMSE, CASI, MoCA, CDR-SB, and IADL (| *r* | = 0.46 ~ 0.55). Performances on the HAI-SAC, SMCB, INF weakly correlated with performances on NPI-SB (|*r*| = 0.11 ~ 0.15) (Table 2).

## Discriminant analyses

**Differentiating MCI from NC.** Table 3 lists performance of screening neuropsychological tests across groups. Fig 1 illustrates discriminative performance of the tests in differentiating cognitive status. HAI-SAC. HAI-SAC (area under the curve, AUC = 0.78) outperformed other screening neuropsychological tests in discriminating MCI from NC (CASI: AUC = 0.72, $\chi^2_{df=1,\ n=761} = 17.42, p < 0.001$; MoCA: AUC = 0.72, $\chi^2_{df=1,\ n=761} = 14.86, p < 0.001$; MMSE: AUC = 0.71, $\chi^2_{df=1,\ n=761} = 20.70, p < 0.001$). In discriminating MCI from NC, a combination of HAI-SAC-3 and HAI-SAC-I outperformed individual elements (HAI-SAC vs. HAI-SAC-3 [AUC = 0.69]: $\chi^2_{df=1,\ n=761} = 75.44, p < 0.001$; HAI-SAC vs. HAI-SAC-I [AUC = 0.72]: $\chi^2_{df=1,\ n=761} = 41.84, p < 0.001$). CASI ($\chi^2_{df=1,\ n=761} = 8.67, p < 0.01$) and MoCA ($\chi^2_{df=1,\ n=761} = 10.13, p < 0.01$) outperformed HAI-SAC-3 but not HAI-SAC-I (CASI: $p = 0.62$; MoCA: $p = 0.79$) in discriminating MCI from NC. In differentiating between MCI and NC, sensitivity of HAI-SAC was 0.87 and specificity was 0.58, with a cut-off score of 41/42.

**Differentiating DAT from MCI.** Performance of HAI-SAC's in discriminating DAT from MCI was on par with performances of CASI, MoCA, and MMSE (AUC: HAI-SAC = 0.87,

**Table 3. Performance on neuropsychological tests across groups.**

|  | CU | MCI | DAT | Statistical significance |
|---|---|---|---|---|
| CASI (maximum = 100) | 83.86 (8.64)[ab] | 73.12 (15.93)[ac] | 44.83 (22.70)[bc] | $p < 0.001$ |
| MMSE (maximum = 30) | 25.81 (3.30)[ab] | 22.24 (5.20)[ac] | 13.98 (6.55)[bc] | $p < 0.001$ |
| MoCA (maximum = 30) | 21.07 (5.10)[ab] | 15.99 (6.33)[ac] | 7.21 (5.40)[bc] | $p < 0.001$ |
| HAI-SAC (maximum = 60) | 48.47 (6.58)[ab] | 38.94 (10.53)[ac] | 21.51 (11.29)[bc] | $p < 0.001$ |
| HAI-SAC-3 (maximum = 40) | 33.95 (4.53)[ab] | 29.77 (6.94)[ac] | 19.42 (9.27)[bc] | $p < 0.001$ |
| CNOCD (maximum = 16) | 14.85 (1.69)[ab] | 13.88 (2.32)[ac] | 10.54 (4.37)[bc] | $p < 0.001$ |
| ML (maximum = 12) | 9.04 (3.07)[ab] | 6.83 (4.17)[ac] | 2.51 (3.55)[bc] | $p < 0.001$ |
| RFO (maximum = 12) | 10.06 (1.57)[ab] | 9.06 (2.07)[ac] | 6.37 (3.29)[bc] | $p < 0.001$ |
| HAI-SAC-I (maximum = 20) | 14.53 (4.19)[ab] | 9.18 (6.36)[ac] | 2.09 (4.71)[bc] | $p < 0.001$ |
| CD (maximum = 8) | 2.71 (3.80)[ab] | 0.76 (2.35)[ac] | 0.12 (0.98)[bc] | $p < 0.001$ |
| FD (maximum = 12) | 7.79 (0.55)[ab] | 6.58 (2.03)[ac] | 2.20 (2.53)[bc] | $p < 0.001$ |

[a] CU ≠ MCI;

[b] CU ≠ DAT;

[c] MCI ≠ DAT.

CD: Cognitive decline reported by informants; CNOCD: Confrontational naming of objects, colors, and details; FD: Functional decline reported by informants; HAI-SAC-3: Objective memory tests of SAC; HAI-SAC-I: Memory and daily functioning gathered from informants; ML: Memory for location; RFO: Recalling functional properties of objects. Other abbreviations are the same as those used in Tables 1 and 2.

CASI = 0.86, MoCA = 0.85; MMSE = 0.85; $p = 0.11$–$0.24$). HAI-SAC and CASI, MMSE, and MoCA outperformed individual HAI-SAC elements in differentiating DAT from MCI (HAI-SAC-3 [AUC = 0.80]: HAI-SAC, $\chi^2_{df=1,\ n=761} = 54,98$, $p < 0.001$; CASI, $\chi^2_{df=1,\ n=761} = 27.24$, $p < 0.001$; MoCA, $\chi^2_{df=1,\ n=761} = 16.69$, $p < 0.001$; MMSE, $\chi^2_{df=1,\ n=761} = 14.46$, $p < 0.001$; HAI-SAC-I [AUC = 0.77]: HAI-SAC, $\chi^2_{df=1,\ n=761} = 53.94$, $p < 0.001$; CASI, $\chi^2_{df=1,\ n=761} = 22.27$, $p < 0.001$; MoCA, $\chi^2_{df=1,\ n=761} = 17.61$, $p < 0.001$; MMSE, $\chi^2_{df=1,\ n=761} = 16.90$, $p < 0.001$). In differentiating between MCI and DAT, sensitivity of HAI-SAC was 0.78 and specificity was 0.84, with a cut-off score of 33/34.

**Differentiating DAT from CU.** In discriminating DAT from CU, HAI-SAC (AUC = 0.98) outperformed CASI, MMSE, and MoCA (CASI [AUC = 0.96]: $\chi^2_{df=1,\ n=761} = 11.32$, $p < 0.001$; MoCA [AUC = 0.96]: $\chi^2_{df=1,\ n=761} = 12.33$, $p < 0.001$; MMSE

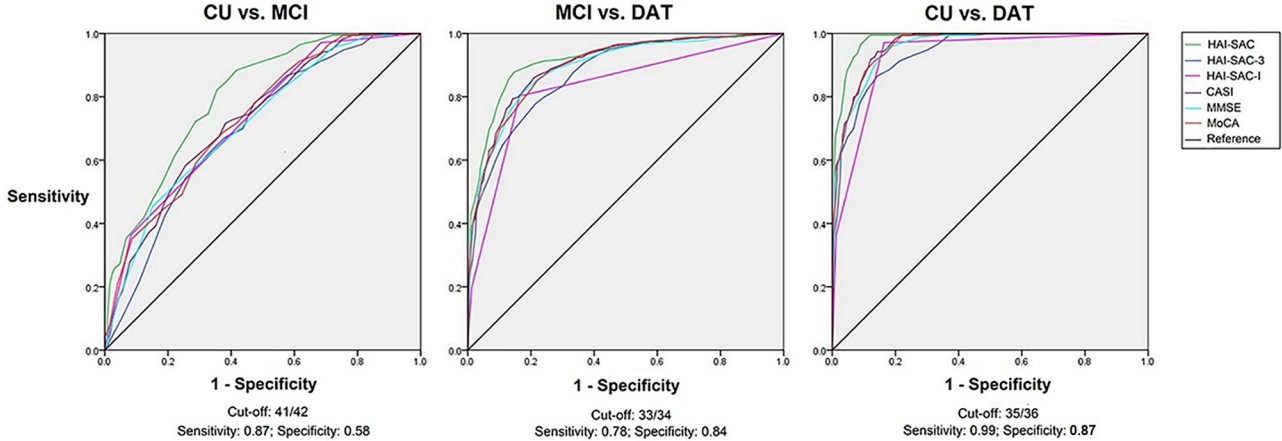

**Fig 1. ROC curves across the screening neuropsychological tests.** Abbreviations are the same as those used in Table 3.

**Table 4. AUC and cut-off points of the assessments.**

|  | HAI-SAC | HAI-SAC-3 | HAI-SAC-I | CASI | MMSE | MoCA |
|---|---|---|---|---|---|---|
| *CU vs. MCI* |  |  |  |  |  |  |
| AUC | 0.78[abcde] | 0.69[afh] | 0.72[bgi] | 0.72[cfg] | 0.71[d] | 0.72[ehi] |
| 95% CI | 0.73–0.83 | 0.63–0.74 | 0.66–0.77 | 0.66–0.77 | 0.66–0.77 | 0.67–0.77 |
| Optimal cut-off | 41/42 | 29/30 | 10 | 81/82 | 27/28 | 15/16 |
| Sensitivity | 0.87 | 0.83 | 0.99 | 0.65 | 0.39 | 0.85 |
| Specificity | 0.58 | 0.46 | 0.28 | 0.66 | 0.90 | 0.46 |
| *MCI vs. DAT* |  |  |  |  |  |  |
| AUC | 0.87[ab] | 0.82[a] | 0.77[b] | 0.86 | 0.85 | 0.85 |
| 95% CI | 0.84–0.90 | 0.79–0.86 | 0.73–0.81 | 0.82–0.89 | 0.81–0.88 | 0.82–0.88 |
| Optimal cut-off | 33/34 | 22/23 | 10 | 65/66 | 19/20 | 10/11 |
| Sensitivity | 0.78 | 0.87 | 0.70 | 0.75 | 0.76 | 0.79 |
| Specificity | 0.84 | 0.60 | 0.84 | 0.81 | 0.79 | 0.72 |
| *CU vs. DAT* |  |  |  |  |  |  |
| AUC | 0.98[abcde] | 0.93[afhj] | 0.93[b] | 0.96[cf] | 0.95[dj] | 0.96[eh] |
| 95% CI | 0.97–0.99 | 0.91–0.95 | 0.91–0.96 | 0.95–0.98 | 0.94–0.97 | 0.95–0.97 |
| Optimal cut-off | 35/36 | 29/30 | 10 | 70/71 | 20/21 | 11/12 |
| Sensitivity | 0.99 | 0.83 | 0.99 | 0.92 | 0.94 | 0.99 |
| Specificity | 0.87 | 0.87 | 0.84 | 0.87 | 0.83 | 0.76 |

[a]: HAI-SAC ≠ HAI-SAC-3;

[b]: HAI-SAC ≠ HAI-SAC-I;

[c]: HAI-SAC ≠ CASI;

[d]: HAI-SAC ≠ MMSE;

[e]: HAI-SAC ≠ MoCA;

[f]: CASI ≠ HAI-SAC-3;

[g]: CASI ≠ HAI-SAC-I;

[h]: MoCA ≠ HAI-SAC-3;

[i]: MoCA ≠ HAI-SAC-I;

[j]: MMSE ≠ HAI-SAC-3.

AUC: Area under the curve; CI: Confidence interval. Other abbreviations are the same as those used in Tables 1 and 2.

[AUC = 0.95]: 15.08, $p < 0.001$) and individual elements of HAI-SAC (HAI-SAC-3 [AUC = 0.92]: $\chi^2_{df=1,\ n=761} = 58.10$, $p < 0.001$; HAI-SAC-I [$p = 0.95$]: $\chi^2_{df=1,\ n=761} = 44.32$, $p < 0.001$). In discriminating between CU and DAT, CASI, MMSE, and MoCA outperformed HAI-SAC-3 (CASI: $\chi^2_{df=1,\ n=761} = 17.16$, $p < 0.001$; MoCA: $\chi^2_{df=1,\ n=761} = 14.00$, $p < 0.001$; MMSE: $\chi^2_{df=1,\ n=761} = 8.97$, $p < 0.01$) but not HAI-SAC-I (CASI: $p = 0.12$; MoCA: $p = 0.15$; MMSE: $p = 0.30$). In differentiating between NC and DAT, sensitivity of HAI-SAC was 0.99 and specificity was 0.89, with a cut-off score of 35/36. Table 4 lists the performance of the SNPAs in discriminating the cognitive status of participants.

## Test time

Time required to complete SNPAs was available among 72% of participants (550 individuals) due to errors of staffs. Average time required to complete HAI-SAC was 4.49 ± 2.30 minutes, which is significantly shorter than time required to complete CASI (17.70 ± 7.08 minutes) or MoCA (11.14 ± 5.93 minutes) ($F_{(2,548)} = 2857.94$, $p < 0.001$) (Fig 2).

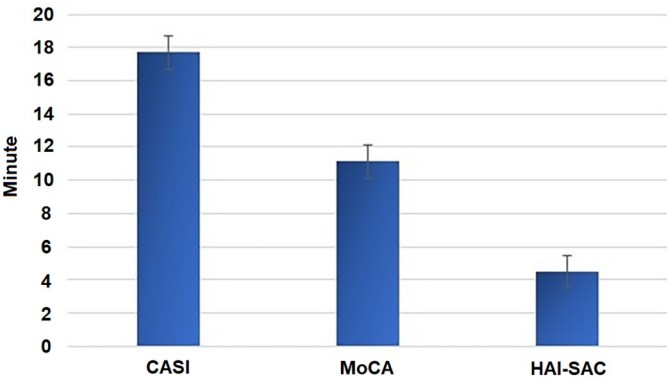

**Fig 2. Test time of the screening neuropsychological tests.**

## Discussion

In the current study, we developed a simple screening method, HAI-SAC, incorporating objective memory assessment and information related to memory functioning gathered from informants to facilitate the determination of cognitive status among individuals with CU, MCI, or DAT. Items of HAI-SAC displayed acceptable internal consistency and can be explained by two factors: Semantic memory and contextual binding and memory functioning gathered from the informants. Performance on the HAI-SAC highly correlated with performances on conventional screening neuropsychological tests included in this study (i.e., CASI, MoCA, and MMSE). Performing HAI-SAC only required less than a quarter and half of the time performing CASI or MoCA, respectively. HAI-SAC outperformed other screening neuropsychological tests included in this study in differentiating MCI from CU with a high degree of sensitivity. Ability of HAI-SAC to differentiate DAT from MCI was on par with CASI. HAI-SAC was more accurate than conventional screening neuropsychological tests included in this study in differentiating DAT from CU.

Recent studies have suggested that value of objective measures of cognitive function in predicting cognitive changes in patients in early stages of AD is strengthened when combined with informant-confirmed memory complaints [10, 12, 14, 16]. The HAI-SAC was intentionally designed to combine the two aspects of data. The moderate internal consistency of HAI-SAC found in the study may be attributable to clear factor structures comprised by objective memory functions and memory functioning gathered from the informants. Thus, HAI-SAC contains assessments of objective memory function and informant-based memory performance. The combination of objective and informant-based assessments of memory functioning of the HAI-SAC may facilitate the early detection of cognitive changes in AD.

It is possible that when combined with informant-based cognitive information (HAI-SAC-I), the HAI-SAC-3 are sufficient to capture the effects of semantic memory and contextual binding deficits on daily functioning. This supposition is supported by the results of EFA, which revealed cross-loadings of FD on both elements of HAI-SAC. It is also possible that combining assessments of core cognitive deficits with functional impacts of deficits could enhance detection of AD by helping to avoid heterogeneity of cognitive changes associated with the isolated medial temporal lobes changes (memory changes with no or a slow progression of functional impairment) [46] or functional cognitive disorder (impaired performance on memory tasks with no or a mild functional impairment confirmed by caregivers) that may not attributable to AD [47–49].

Efficiency in the diagnosis of cognitive status in AD patients is often a critical issue in clinical settings [5, 9]. Comprehensive neuropsychological assessments are not always feasible [2, 3]. Thus, an efficient screening neuropsychological test is needed to enhance case identification in AD. Our results revealed that HAI-SAC required less than a quarter of the time completing CASI and half of the time completing MoCA while providing higher accuracy in discriminating between MCI and CU and between DAT and CU. Thus, HAI-SAC is a valuable and efficient option for cognitive assessment in AD.

One of the challenges in differentiation between CU and MCI involves demarcation between effects of normal aging and pathological changes [50, 51]. For example, changes in executive functions and working memory are common among normal elderly individuals [50]. HAI-SAC provides a high degree of sensitivity but relatively low specificity in differentiating MCI from CU, which may be attributable to cognitive complaints to which normal elderly individuals are prone [52]. Our results also revealed low CD scores in CU group. High FD scores in the CU group might reflect the effects of cognitive reserve among some CU individuals [53]. For example, individuals with higher educational levels may possess superior resilience under the effects of AD-related neurological changes [54]. Similarly, the gradient observed in education level from the CU group to the MCI and DAT groups might also reflect the effects of cognitive reserve on cognitive progression.

Conventional screening neuropsychological tests are highly sensitive to education levels of participants [7]. In the current study, we performed *post-hoc* analysis of the relationships between education levels and SNPA. Our results revealed that of correlation between education level and conventional SNPA scores was higher than correlation between education level and HAI-SAC (Pearson's $r$: HAI-SAC: 0.36, vs. CASI [$r = 0.46$]: $Z = -2.20$, $p < 0.03$, MoCA [$r = 0.53$]: $Z = -4.16$, $p < 0.001$, MMSE [$r = 0.48$]: $Z = -2.72$, $p < 0.01$). Thus, the conventional SNPA may be more suitable in assessing the level of cognitive reserve [54], particularly when dealing with individuals with a high education level. By contrast, HAI-SAC may be more sensitive in detecting a breakdown of cognitive reserve [55]. This may also explain why the sensitivity of conventional screening neuropsychological tests is lower than that of HAI-SAC.

Previous studies have suggested that executive functions could be an important index in predicting cognitive changes in patients with MCI [56]. Performance of CASI in differentiating DAT and MCI was comparable to that of HAI-SAC. It appears that CASI and MoCA place a heavy cognitive load on executive functions (e.g., problem-solving, mental manipulation, cognitive fluency, switching, and inhibitory control), exceeding that of MMSE [7, 8]. In contrast, HAI-SAC focuses mainly on core memory problems in early-stage AD patients, and in so doing discriminative power in differentiating DAT from MCI. Thus, it is likely that patterns in memory deficits and executive dysfunctions attributable to AD could both be used to differentiate between MCI and DAT [57–59]. In discriminating between DAT and MCI, the performance of MMSE was on par with that of CASI and MoCA. A low score on MMSE reflects a global decline in cognition in cases of DAT.

In discriminating between MCI and CU, the performance of MMSE was exceeded by that of HAI-SAC; however, it was on par with that of CASI and MoCA. In differentiating MCI from CU, the sensitivity of MMSE was low (0.39) but specificity was high (0.90). In the future, researchers could perhaps combine HAI-SAC and MMSE to balance sensitivity and specificity in differentiating between CU and MCI.

It is possible that patients with MCI are able to use cognitive strategies to compensate for memory deficits [59]. By contrast, it appears that cognitive deficits in patients with DAT interfere with cognitive compensation [60]. Thus, it appears that the association between objective memory loss and functional deficits in daily activities is an important issue in assessing DAT. In this study, conventional screening neuropsychological tests performed well in

differentiating between DAT and CU (AUC = 0.92 ~ 0.95), due perhaps to their efficacy in capturing severe global changes in cognition. Accuracy of HAI-SAC in discriminating DAT from CU (AUC = 0.98) exceeded that of conventional screening neuropsychological tests. It therefore supports the idea that association between objective memory loss and functional changes in daily activities is important to differentiating between DAT and CU.

The screening neuropsychological test proposed in this study proved to be effective in determining cognitive status of patients with AD. Our results also shed light on nature of cognitive and functional deficits in patients with AD. Nonetheless, this study was subject to a number of limitations. First, diagnosis of cognitive status was based on clinical information that included performance on CASI. It is likely therefore that actual discriminative power of CASI in determining cognitive status of AD patients may have been overestimated. Second, based on cross-sectional nature of study, we could not exclude possibility that some participants in CU group were in fact in preclinical stages of AD [4]. In addition, the predictive values of HAI-SAC score on cognitive decline is unclear. Future studies should adopt longitudinal design and investigate value of HAI-SAC in discriminating individuals undergoing normal aging and those in preclinical AD. Third, limitations on sample size necessitated uniform cut-off scores in differentiating cognitive status of patients with AD. Note that uniform cut-off scores can hinder detection of cognitive changes in heterogeneous populations. Fourth, the diagnosis of MCI rely on the global score of CASI and thus the MCI patients may not be in the early stages of MCI (e.g., amnestic mild cognitive impairment-single domain, aMCI-sd) [61]. Future studies should include patients with aMCI-sd or preclinical AD and investigate the discriminative abilities of HAI-SAC among individuals with normal and early pathological aging. Fifth, the fact that the MMSE score was extracted from CASI in this study may explain the lack of a significant performance difference between MMSE and CASI and MoCA in discriminating cognitive status. Sixth, we found that the number of individuals with hypercholesterolemia in the CU and MCI groups exceeded that in the DAT group. Previous studies have suggested that cholesterol has potentially protective effects on the cognition of elderly individuals [62]; however, that assertion and our findings to support it will require further validation. Finally, the trade-off of sensitivity and specificity is important in clinical evaluation of individuals suspected having pathological aging. Future study should evaluate the false positive rate of HAI-SAC performance using longitudinal design.

Our results provide evidence that the proposed HAI-SAC is a valid, concise, and efficient neuropsychological assessment by which to determine the cognitive status of AD patients. Benefits of the proposed instrument may be attributable to the fact that it addresses early functional impact of core memory decline. Future studies should investigate effectiveness of HAI-SAC in predicting longitudinally cognitive decline.

## Acknowledgments

We thank members in the HAICDDS for their help on this study.

## Author Contributions

**Conceptualization:** Hsin-Te Chang, Pai-Yi Chiu.

**Data curation:** Hsin-Te Chang, Pai-Yi Chiu.

**Formal analysis:** Hsin-Te Chang, Pai-Yi Chiu.

**Investigation:** Hsin-Te Chang, Pai-Yi Chiu.

**Methodology:** Hsin-Te Chang, Pai-Yi Chiu.

**Project administration:** Pai-Yi Chiu.

**Resources:** Pai-Yi Chiu.

**Software:** Hsin-Te Chang.

**Supervision:** Pai-Yi Chiu.

**Validation:** Pai-Yi Chiu.

**Visualization:** Hsin-Te Chang.

**Writing – original draft:** Hsin-Te Chang.

**Writing – review & editing:** Hsin-Te Chang, Pai-Yi Chiu.

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
