## [Decision Letter · Decision Letter 0]

26 Oct 2022

PONE-D-22-23424Development of a Simple Screening Tool for Determining Cognitive Status in Alzheimer’ DiseasePLOS ONE

Dear Dr. Chiu,

Thank you for submitting your manuscript to PLOS ONE. After careful consideration, we feel that it has merit but does not fully meet PLOS ONE’s publication criteria as it currently stands. Therefore, we invite you to submit a revised version of the manuscript that addresses the points raised during the review process. Please submit your revised manuscript by Dec 10 2022 11:59PM. If you will need more time than this to complete your revisions, please reply to this message or contact the journal office at plosone@plos.org. Please include the following items when submitting your revised manuscript:A rebuttal letter that responds to each point raised by the academic editor and reviewer(s). You should upload this letter as a separate file labeled 'Response to Reviewers'.A marked-up copy of your manuscript that highlights changes made to the original version. You should upload this as a separate file labeled 'Revised Manuscript with Track Changes'.An unmarked version of your revised paper without tracked changes. You should upload this as a separate file labeled 'Manuscript'.If applicable, we recommend that you deposit your laboratory protocols in protocols.io to enhance the reproducibility of your results. Protocols.io assigns your protocol its own identifier (DOI) so that it can be cited independently in the future. For instructions see: https://journals.plos.org/plosone/s/submission-guidelines#loc-laboratory-protocols. Additionally, PLOS ONE offers an option for publishing peer-reviewed Lab Protocol articles, which describe protocols hosted on protocols.io. Read more information on sharing protocols at https://plos.org/protocols?utm_medium=editorial-email&utm_source=authorletters&utm_campaign=protocols.

We look forward to receiving your revised manuscript.

Kind regards,

Alessandra Coin

Academic Editor

PLOS ONE

Journal Requirements:

Additional Editor Comments:

O agree with both reviewers for a minor revision of the manuscript and I think they raised interesting points to imporve the quality of yoou manuscript.

Reviewers' comments:

Reviewer's Responses to Questions

**Comments to the Author**

1. Is the manuscript technically sound, and do the data support the conclusions?

Reviewer #1: Yes

Reviewer #2: Yes

2. Has the statistical analysis been performed appropriately and rigorously? 

Reviewer #1: Yes

Reviewer #2: Yes

3. Have the authors made all data underlying the findings in their manuscript fully available?

Reviewer #1: No

Reviewer #2: Yes

4. Is the manuscript presented in an intelligible fashion and written in standard English?

Reviewer #1: Yes

Reviewer #2: Yes

5. Review Comments to the Author

Reviewer #1: Authors propose an interesting and easy-to-administer tool for the detection of cognitive impairment and, even more interestingly, for differentiating between MCI and dementia. I certainly encourage this paper to be published because of its likely utility in daily clinical practice.

I only have one minor comment that I would like the authors may discuss. Although they mentioned educational level and, as a matter of fact, they did not find an association with the pathology evolution, in such a study, I would expect some considerations about cognitive reserve and its role in modulating not only the onset, but also the progression of cognitive impairment.

Furthermore, although MMSE has important limitations (clinical, first of all!), it would be helpful also a comparison/a critical consideration or comment referring HAI-SAC to MMSE which is, at the date, the most applied screening test for dementia (authors certainly know that MoCA is more difficult and more focused on executive functions rather than MMSE which is easier and, for some aspects, more suitable for older individuals).

Reviewer #2: Dear Editor, Thank you for your invitation.

ABSTRACT should be re-written in order to provide clearer information to the readers.

Please provide the mean age and female percentage of the study population. How many patients were diagnosed with MCI, AD and control group? What was the sensitivity, specifity of the HAI-SAC to detect MCI and AD? And it would be better ig the authors mention the AUC results about discriminating MCI from NC/AD.

INTRO: The authors should give some examples about short screening tools (https://pubmed.ncbi.nlm.nih.gov/28660847/, https://pubmed.ncbi.nlm.nih.gov/29923472/)

RESULTS is well-written, but is there any information about comorbidities of the patients?

6. PLOS authors have the option to publish the peer review history of their article (what does this mean?). If published, this will include your full peer review and any attached files.

Reviewer #1: **Yes: **Maria Deita

Reviewer #2: No

---

## [Author Response · Author response to Decision Letter 0]

23 Nov 2022

PLOS ONE

Referee Report on the Revised Version of Manuscript PONE-D-22-23424

“Development of a Simple Screening Tool for Determining Cognitive Status in Alzheimer’s Disease”

We are grateful for your invaluable assistance in the revision of this paper. We have closely followed the guidance provided by the reviewers and editors in this revision. Detailed responses to the comments are provided below.

Journal Requirements

Reply: 

The formatting of the revised manuscript has been adjusted in accordance with journal requirements.

2. Please provide additional details regarding participant consent. In the ethics statement in the Methods and online submission information, please ensure that you have specified (1) whether consent was informed and (2) what type you obtained (for instance, written or verbal, and if verbal, how it was documented and witnessed). If your study included minors, state whether you obtained consent from parents or guardians. If the need for consent was waived by the ethics committee, please include this information. If you are reporting a retrospective study of medical records or archived samples, please ensure that you have discussed whether all data were fully anonymized before you accessed them and/or whether the IRB or ethics committee waived the requirement for informed consent. If patients provided informed written consent to have data from their medical records used in research, please include this information. Once you have amended this/these statement(s) in the Methods section of the manuscript, please add the same text to the “Ethics Statement” field of the submission form (via “Edit Submission”). For additional information about PLOS ONE ethical requirements for human subjects research, please refer to http://journals.plos.org/plosone/s/submission-guidelines#loc-human-subjects-research.

Reply:

As the data were analyzed retrospectively and anonymously from a database established in previous Taiwanese studies, informed consent was waived. We have provided the information in the Methods section of the revised manuscript (Lines 12 to 15, Page 7).

Reply:

We have uploaded a de-identifying minimum dataset that can support the conclusions of the manuscript on BioStudies Submission Tool (accession number: S-BSST951). We have added a section (Data Availability) in the revised manuscript (Lines 14 to 15, Page 24).

Reply:

We have corrected and updated the reference list in the revised manuscript (Page 25 to 35).

5. Additional Editor Comments: I agree with both reviewers for a minor revision of the manuscript and I think they raised interesting points to improve the quality of your manuscript.

Reply:

We have closely followed the guidance provided by the reviewers in this revision.

Reviewer 1

Authors propose an interesting and easy-to-administer tool for the detection of cognitive impairment and, even more interestingly, for differentiating between MCI and dementia. I certainly encourage this paper to be published because of its likely utility in daily clinical practice.

1. I only have one minor comment that I would like the authors may discuss. Although they mentioned educational level and, as a matter of fact, they did not find an association with the pathology evolution, in such a study, I would expect some considerations about cognitive reserve and its role in modulating not only the onset, but also the progression of cognitive impairment.

Reply:

We have addressed the issue of cognitive reserve in the Discussion section of the revised manuscript (Lines 10 to 14, Page 14 and Line 20, Page 21 to Line 1, Page 22).

2. Furthermore, although MMSE has important limitations (clinical, first of all!), it would be helpful also a comparison/a critical consideration or comment referring HAI-SAC to MMSE which is, at the date, the most applied screening test for dementia (authors certainly know that MoCA is more difficult and more focused on executive functions rather than MMSE which is easier and, for some aspects, more suitable for older individuals).

Reply:

We have added a comparison and addressed the sensitivity and specificity of HAI-SAC and MMSE in the Discussion section of the revised manuscript (Lines 11 to 18, Page 22). We have also mentioned the limitation imposed by our use of an extracted MMSE score (Lines 21 to 23, Page 23).

Reviewer 2

1. ABSTRACT should be re-written in order to provide clearer information to the readers.

Reply:

We have re-written the Abstract in the revised manuscript (Page 3).

2. Please provide the mean age and female percentage of the study population.

Reply:

We have provided this information in the Results (Lines 12 to 14, Page 13) and Abstract (Lines 5 to 6, Page 3) section of the revised manuscript.

3. How many patients were diagnosed with MCI, AD and control group?

Reply:

A total of 397 participants were diagnosed with AD dementia, 231 were diagnosed with MCI, and 136 were diagnosed as CU. This information has been included in the Methods section (Lines 10 to 12, Page 7) and Table 1 of the revised manuscript. We have also provided this information in the Abstract section of the revised manuscript (Lines 6 to 7, Page 3).

4. What was the sensitivity, specificity of the HAI-SAC to detect MCI and AD?

Reply:

In differentiating between MCI and NC, the sensitivity of HAI-SAC was 0.87 and the specificity was 0.58, with a cut-off score of 41/42. In differentiating between MCI and DAT, the sensitivity of HAI-SAC was 0.78 and the specificity was 0.84, with a cut-off score of 33/34. In differentiating between NC and DAT, the sensitivity of HAI-SAC was 0.99 and the specificity was 0.89, with a cut-off score of 35/36. This information has been included in the Results section of the revised manuscript (Lines 2 to 11, Page 16; Lines 7 to 14 and Lines 18 to 21, Page 17). We have also provided the information in the Abstract section of the revised manuscript (Line 19, Page 3 to Line 1, Page 4) and corrected typos in Table 4 (Page 19).

5. And it would be better if the authors mention the AUC results about discriminating MCI from NC/AD.

Reply:

This information has been included in the Results section and Table 4 of the revised manuscript (Pages 18 to 19). We have also included this information in the Abstract section of the revised manuscript (Line 22, Page 3 to Line 1, Page 4).

6. INTRO: The authors should give some examples about short screening tools (https://pubmed.ncbi.nlm.nih.gov/28660847/, https://pubmed.ncbi.nlm.nih.gov/29923472/)

Reply:

We have included an introduction to these tools in the Introduction section of the revised manuscript (Lines 13 to 15, Page 5).

7. RESULTS is well-written, but is there any information about comorbidities of the patients?

Reply:

We have listed the proportion of individuals with hypertension, diabetes, hypercholesterolemia, or coronary artery disease across groups in the Methods (Lines 6 to 8, Page 7) and Results (Lines 1 to 4, Page 14) sections and Table 1 of the revised manuscript (Page 14).

---

## [Decision Letter · Decision Letter 1]

22 Dec 2022

Development of a Simple Screening Tool for Determining Cognitive Status in Alzheimer’s Disease

PONE-D-22-23424R1

Dear Dr. Chiu,

We’re pleased to inform you that your manuscript has been judged scientifically suitable for publication and will be formally accepted for publication once it meets all outstanding technical requirements.

Kind regards,

Alessandra Coin

Academic Editor

PLOS ONE

Additional Editor Comments (optional):

Reviewers' comments:

Reviewer's Responses to Questions

**Comments to the Author**

1. If the authors have adequately addressed your comments raised in a previous round of review and you feel that this manuscript is now acceptable for publication, you may indicate that here to bypass the “Comments to the Author” section, enter your conflict of interest statement in the “Confidential to Editor” section, and submit your "Accept" recommendation.

Reviewer #1: All comments have been addressed

Reviewer #2: All comments have been addressed

2. Is the manuscript technically sound, and do the data support the conclusions?

Reviewer #1: Yes

Reviewer #2: Yes

3. Has the statistical analysis been performed appropriately and rigorously? 

Reviewer #1: Yes

Reviewer #2: Yes

4. Have the authors made all data underlying the findings in their manuscript fully available?

Reviewer #1: No

Reviewer #2: Yes

5. Is the manuscript presented in an intelligible fashion and written in standard English?

Reviewer #1: Yes

Reviewer #2: Yes

6. Review Comments to the Author

Reviewer #1: (No Response)

Reviewer #2: (No Response)

7. PLOS authors have the option to publish the peer review history of their article (what does this mean?). If published, this will include your full peer review and any attached files.

Reviewer #1: **Yes: **Maria Devita

Reviewer #2: **Yes: **Pinar Soysal

---

## [Editor Report · Acceptance letter]

3 Jan 2023

PONE-D-22-23424R1 

Development of a simple screening tool for determining cognitive status in Alzheimer’s disease 

Dear Dr. Chiu:

I'm pleased to inform you that your manuscript has been deemed suitable for publication in PLOS ONE. Congratulations! Your manuscript is now with our production department. 

Kind regards, 

on behalf of

Dr. Alessandra Coin 

Academic Editor

PLOS ONE